# Neural Field Classifiers via Target Encoding and Classification Loss

**Xindi Yang**$^{\mu,\gamma}\star$   **Zeke Xie**$^{\gamma}\star\dagger$
**Xiong Zhou**$^{\gamma}$   **Boyu Liu**$^{\gamma}$   **Buhua Liu**$^{\gamma}$
**Yi Liu**$^{\mu}$   **Haoran Wang**$^{\gamma}$   **Yunfeng Cai**$^{\gamma}$   **Mingming Sun**$^{\gamma}$
$^{\mu}$Beijing Key Lab of Traffic Data Analysis and Mining, Beijing Jiaotong University
$^{\gamma}$Cognitive Computing Lab, Baidu Research

## Abstract

Neural field methods have seen great progress in various long-standing tasks in computer vision and computer graphics, including novel view synthesis and geometry reconstruction. As existing neural field methods try to predict some coordinate-based continuous target values, such as RGB for Neural Radiance Field (NeRF), all of these methods are regression models and are optimized by some regression loss. However, are regression models really better than classification models for neural field methods? In this work, we try to visit this very fundamental but overlooked question for neural fields from a machine learning perspective. We successfully propose a novel Neural Field Classifier (NFC) framework which formulates existing neural field methods as classification tasks rather than regression tasks. The proposed NFC can easily transform arbitrary Neural Field Regressor (NFR) into its classification variant via employing a novel Target Encoding module and optimizing a classification loss. By encoding a continuous regression target into a high-dimensional discrete encoding, we naturally formulate a multi-label classification task. Extensive experiments demonstrate the impressive effectiveness of NFC at the nearly free extra computational costs. Moreover, NFC also shows robustness to sparse inputs, corrupted images, and dynamic scenes.

## 1 Introduction

**Background** Neural field methods emerge as promising methods for parameterizing a field, represented by a scalar, vector, or tensor, that has a target value for each point in space and time. Neural field methods first gained great attention in computer vision and computer graphics, because learning-based neural field methods show impressive performance in novel view synthesis and surface reconstruction. Synthesizing novel-view images of a 3D scene from a group of images is a long-standing task (Chen and Williams, 1993; Debevec et al., 1996; Levoy and Hanrahan, 1996; Gortler et al., 1996; Shum and Kang, 2000) and has recently made significant progress with Neural Radiance Field (NeRF) (Liu et al., 2020; Mildenhall et al., 2021). Neural surface representation (Michalkiewicz et al., 2019; Niemeyer et al., 2020; Yariv et al., 2021; Wang et al., 2021) is another important and long-standing problem orthogonal to novel view synthesis.

**NeRF Basics** Without losing generality, we take a standard NeRF as the example. NeRF can efficiently represent a given scene by implicitly encoding volumetric density and color through a coordinate-based neural network (often referred to as a simple multilayer perceptron or MLP). NeRF regresses from a single 5D representation $(x, y, z, \theta, \phi)$- 3D coordinates $\boldsymbol{x} = (x, y, z)$ plus 2D viewing directions $\boldsymbol{d} = (\theta, \phi)$- to a single volume density $\sigma$ and a view-dependent color $\boldsymbol{c} = (r, g, b)$. NeRF approximates this continuous 5D scene representation with an MLP network $f_\Theta : (\boldsymbol{x}; \boldsymbol{d}) \rightarrow (\boldsymbol{c}; \sigma)$ and optimizes its weights $\Theta$ to map each input 5D coordinate to the corresponding volume density and directional emitted color.

For a target view with pose, a camera ray can be parameterized as $\boldsymbol{r}(t) = \boldsymbol{o} + t\boldsymbol{d}$ with the ray origin $\boldsymbol{o}$ and ray unit direction $\boldsymbol{d}$. The expected color $\boldsymbol{C}(\boldsymbol{r})$ of camera ray $\boldsymbol{r}(t)$ with near and far bounds $t_n$

---

$\star$ Equal Contributions; $\dagger$ Correspondence to *xiezeke@baidu.com*.
  Code: https://github.com/Madaoer/Neural-Field-Classifier.

and $t_f$ is

$$\hat{\boldsymbol{C}}(\boldsymbol{r}) = \int_{t_n}^{t_f} T(t)\sigma(t)\boldsymbol{c}(t)dt, \tag{1}$$

where $T(t) = \exp(-\int_{t_n}^{t} \sigma(s)ds)$ denotes the accumulated transmittance along the ray from $t_n$ to $t$. For simplicity, we have ignored the coarse and fine renderings via different sampling methods.

The rendered image pixel value for camera ray $\boldsymbol{r}$ can then be compared against the corresponding ground truth pixel color value $\boldsymbol{C}(\boldsymbol{r})$, through the $N$ sampled points along the ray. Note that $\boldsymbol{C} = (R, G, B)$ is the color vector and $R, G, B \in [0, 1]$ are the normalized continuous values, while the raw color values are integers in $[0, 255]$. The conventional rendering loss is the regression loss

$$L(\Theta) = \frac{1}{\|\mathcal{R}\|} \sum_{\boldsymbol{r} \in \mathcal{R}} \|\hat{\boldsymbol{C}}(\boldsymbol{r}) - \boldsymbol{C}(\boldsymbol{r})\|_2^2, \tag{2}$$

where $\|\cdot\|_2$ is the $\ell_2$ norm , the ray/pixel $\boldsymbol{r} = (\boldsymbol{x_r}, \boldsymbol{d_r}, \boldsymbol{c_r})$, and $\mathcal{R}$ is the training data (minibatch).

**Motivation** As the targets are continuous values in previous studies, people naturally formulate neural fields as regression models. In this work, we visit a very fundamental but overlooked question: are regression formulation really better than classification formulation for neural field methods? The answer is negative. There exist overlooked pitfalls in existing neural fields. For example, NeRF and its variants output $N$ points' predictions per pixel. Compared to classical supervised learning methods where each data point has its own ground-truth label, supervision signals for NeRF are obviously very weak and insufficient. In the presence of weak or noisy supervision, neural networks may exhibit significant overfitting and poor generalization (Zhang et al., 2017; Xie et al., 2021a).

**Contributions** We mainly make three contributions.

1. We successfully design a novel Neural Field Classifier (NFC) framework which formulates neural fields as classification tasks rather than regression tasks.

2. We propose Target Encoding and introduce a classification loss to transform existing Neural Field Regressors (NFR) into NFC. The implementation is quite simple, as we only need to revise the final layer of the original NFR and optimize a classification loss.

3. We are the first to explore regression versus classification for neural fields. Surprisingly, classification models significantly and generally outperform its conventional regression counterparts in extensive experiments at the nearly free extra computational cost.

## 2 METHODOLOGY

In this section, we formally introduce the NFC framework and two key ingredients: Target Encoding and classification loss. Figure 1 illustrates the structure of NFC and NFR. .

**Target Encoding and Decoding** We first discuss our Target Encoding module and how to revise existing neural field methods according to the encoding-decoding rule.

There are various ways to encode an continuous value into a discrete vector. For example, NeRF projects the input continuous coordinates into a high-dimensional continuous vector via positional encoding before feeding the inputs into neural networks. Thus, this is a kind of input encoding rule. Our target encoding requires projecting a color value into a discrete vector so that we can classify.

Suppose $\boldsymbol{C} = (R, G, B)$ is the three-channel color. In the raw dataset without preprocessing, the color values are actually integers in $[0, 255]$. For simplicity, we ignore the channel and consider a single color value $y \in [0, 255]$. A very naive target encoding rule is that we directly use $y$ as the class label. Then we quickly formulate a 256-class classification problem via one-hot encoding for each channel. However, this one-hot target encoding rule is naive and inefficient for two reasons. First, the number of logits increases to 768 from 3, which can cost more memory and computational costs than the original simple MLP. Second, this naive target encoding ignore the relevant information carried by the classes. Suppose the ground-truth label of a sample is 0. If a model $A$ predicts 1 and a model $B$ predicts 255, their loss will be equally high. This is obviously unreasonable, because 1 is a much better prediction than 255.

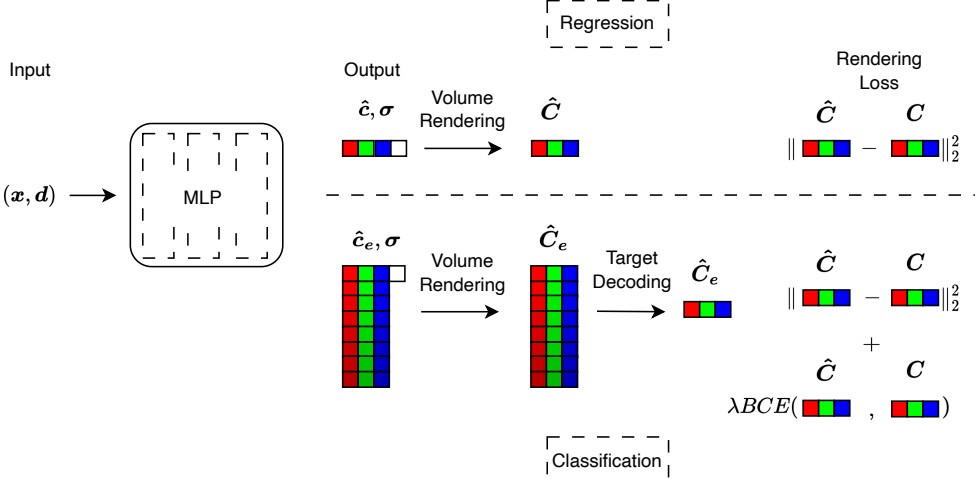

Figure 1: The illustration of standard Neural Field Regressors and our Neural Field Classifiers. Our method makes two modifications on existing neural fields. First, due to the Target Encoding module, the final output of neural networks need to be a high-dimensional color encoding rather than a three-channel color value itself. The designed encoding rule connects our high-dimensional discrete representation and the standard three-channel continuous representation. Second, we mainly use a classification loss as the main optimization objective. Note that the classification loss, as the main optimization objective, can be larger than the standard MSE loss by two orders of magnitude.

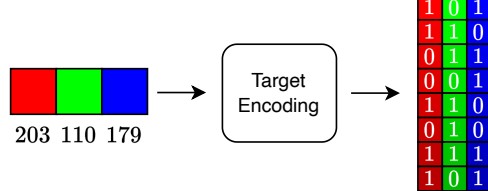

Figure 2: The illustration of Binary-Number Target Encoding.

Thus, a better target encoding rule is desired for neural field methods. Fortunately, we discover that the classical binary-number system can work well as the target encoding rule for NFC. A binary number is a number expressed in the base-2 numeral system, a method of mathematical expression which uses only two symbols: 0 and 1. With such binary-number encoding rule, we can express a color value $y$ as an 8-bit binary number $\boldsymbol{y}$, namely $\boldsymbol{y} = \mathrm{BinaryEncod}(y)$. For example, $\boldsymbol{y} = \mathrm{BinaryEncod}(203) = [1, 1, 0, 0, 1, 0, 1, 1]$ can serve as the label vector for an 8-label binary classification task. We illustrate the binary-number encoding in Figure 2.

Finally, we may let a neural network to predict 0 or 1 for each bit in the label vector $\boldsymbol{y}$ as long as we increase the number of color logits from 3 to 24 and design a proper classification loss.

**Classification Loss** Classification losses are usually some probabilistic losses, such as Cross Entropy (CE). In this work, we also use an CE-based classification loss as the main optimization objective. In the binary-number system, the place values increase by the factor of 2. We also need to let the class weight increase by the factor of 2 as the place values.

We first formulate a bit-wise classification loss as

$$l_{\mathrm{b}}(\hat{\boldsymbol{y}}, \boldsymbol{y}) = \frac{1}{255} \sum_{j=1}^{8} 2^{j-1} \, \mathrm{BCE}(\hat{\boldsymbol{y}}^{(j)}, \boldsymbol{y}^{(j)}), \tag{3}$$

where $j$ is the place index and $\hat{\boldsymbol{y}}$ is the predicted probability given by the final layer of the MLP. We note that $2^{j-1}$ assigns a higher weight to the class with a higher place value.

An alternative choice is that we first decode the predict probability $\hat{y}$ back into a continuous value $\hat{C} = \frac{1}{255} \text{BinaryDecod}(\hat{y})$. We treat $\hat{C}$ as the weighted-averaged predicted probability for a channel and interpret the ground-truth color value ($\in [0, 1]$) as the ground-truth probabilistic soft label. Then we obtain a channel-wise classification loss as

$$l_c(\hat{C}, C) = \text{BCE}(\hat{C}, C) = \text{BCE}\left(\frac{1}{255} \text{BinaryDecod}(\hat{y}), C\right). \qquad (4)$$

Which classification loss should we choose? According to our empirical analysis, we find that two classification losses both significantly improve exist neural fields, while the channel-wise classification loss given by (4) has a simpler implementation than the bit-wise classification loss given by (3). In the following of this paper, we use the channel-wise classification loss as the default classification loss unless we specify it otherwise.

We point out that, due to the process of ray sampling and volume rendering, the predicted probability $\hat{C}$ (or $\hat{y}$) of NFC does not strictly lie in $(0, 1)$ like a standard image classification model. In the case that the predicted probability is greater than or equal to one, the gradient of the classification loss may explode. This case is impossible in image classification but may happen in neural fields due to volume rendering. Thus, we slightly modify the classification loss as

$$l_c(\hat{C}, C) = \text{BCE}(\min(\hat{C}, 1 - \epsilon), C), \qquad (5)$$

where $\epsilon = 0.001$ is desired for the numerical stability purpose.

As the $\min(\cdot, \cdot)$ operation is non-differentiable, for the data points with $C \notin (0, 1)$, the gradient will vanish and does not update neural networks. To solve the gradient vanishing problem, we let the standard Mean Squared Error (MSE) loss serve as a minor optimization objective. So the final optimization objective of NFC can be formulated as follow

$$L_{\text{NFC}}(\hat{C}, C) = \|\hat{C} - C\|_2^2 + \lambda \text{BCE}(\min(\hat{C}, 1 - \epsilon), C), \qquad (6)$$

where $\hat{C} = \frac{1}{255} \text{BinaryDecod}(\hat{y})$ is the predicted color probability/value.

We note that the classification loss term is the main optimization objective, because the BCE-based classification loss can be significantly larger than the standard MSE loss by more than one order of magnitude during almost the whole training process (after the initial tens of iterations). In practice, we do not need to fine-tune the hyperparameter $\epsilon$. Fine-tuning $\lambda$ is easy, because the advantage of NFC over NFR is robust to a wide range of $\lambda$ (e.g. $[0.1, 100]$).

## 3 RELATED WORK

In this section, we review representative related works and discuss their relations to our method.

**Neural Fields** Physicists proposed the concept of fields to continuously parameterize an underlying physical quantity of an object or scene over space and time. Fields have been used to describe physical phenomena (Sabella, 1988) and coordinate-based phenomena beyond physics, such as computing image gradients (Schlüns and Klette, 1997) and simulating collisions (Osher et al., 2004). Recent advances in computer vision and computer graphics showed increased interest in employing coordinate-based neural networks, also called implicit neural networks, that maps a 3D spatial coordinate to a flow field in fluid dynamics, or a colour and density field in 3D scene representation. Such neural networks are often referred to as neural fields (Xie et al., 2022a) or implicit neural representations (Sitzmann et al., 2020; Michalkiewicz et al., 2019; Niemeyer et al., 2020; Yariv et al., 2021; Wang et al., 2021). Our previous work (Xie et al., 2023b) proposed a novel loss but focused on employing structural information, while this work followed the experimental setting. To the best of our knowledge, previous studies did not touch a classification framework of neural fields.

**Target Encoding** Target Encoding, also called Target Embedding, Label Encoding, or Label Embedding, has been studied in previous classification studies of machine learning (Bengio et al., 2010; Akata et al., 2013; 2015; Rodríguez et al., 2018; Jia and Zhang, 2021). However, previous studies only study how to project one discrete-label space into another discrete-label space for standard classification tasks. They failed to explore how to encode continuous targets for regression tasks into discrete targets for classification tasks. The comparisons between regression formulation and classification formulation of one machine learning task is also largely under-explored in related machine learning studies.

## 4 EMPIRICAL ANALYSIS AND DISCUSSION

In this section, we conduct extensive experiments to demonstrate the effectiveness of NFCs over their standard regression counterparts.

We let the experimental settings follow original papers to produce the baselines, unless we specify it otherwise. The basic principle of our experimental settings is to fairly compare NFC and NFR. Thus, we keep all hyperparameters same for them. Our evaluation is also in line with the established image and geometry assessment protocols within neural fields community. More experiments details can be found in Appendix A.

We mainly choose four representative neural field methods as the backbones, including DVGO (Sun et al., 2022), vanilla NeRF (Mildenhall et al., 2021), D-NeRF (Pumarola et al., 2021), and NeuS (Wang et al., 2021). We present more quantitative results and supplementary experimental results of Strivec (Gao et al., 2023) and 4DGS (Wu et al., 2023) in Appendix B. We implement the NFC variants by modifying the final layers' logits of the neural network architectures and the optimization objective. For simplicity, we refer to NFC and NFR of a backbone (e.g. NeRF) as NeRF-R and NeRF-C, respectively.

### 4.1 NOVEL VIEW SYNTHESIS EXPERIMENTS

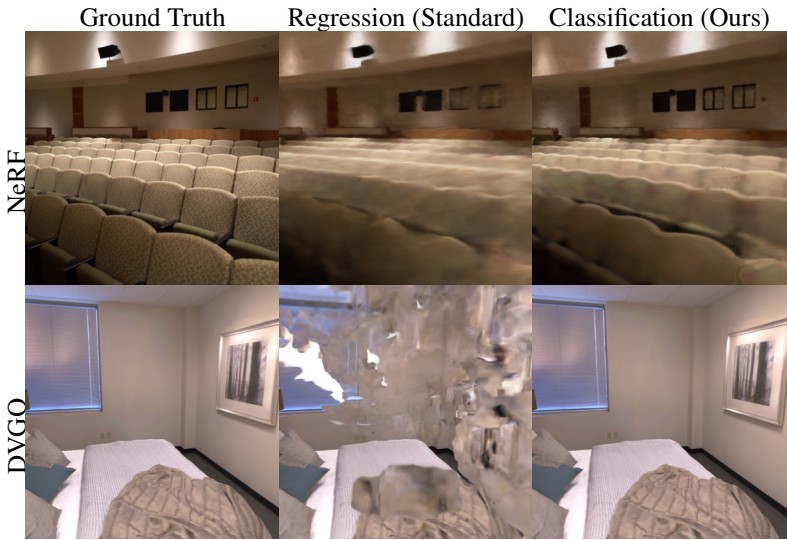

Figure 3: Qualitative comparisons of NFC and NFR for static scenes. Top Row: NeRF. Bottom Row: DVGO.

**Static Scene** We first empirically study novel view synthesis on common static scenes. We choose the Replica Dataset and Tanks and Temples Advanced(T & T) as the benchmark datasets. The Replica Dataset encompasses eight complex static scenes characterized by dense geometry, high resolution HDR textures, and sparse training views. T & T Dataset is a popular 3D reconstruction dataset known for its challenging conditions, including weak illumination, uniform appearance surfaces, and large-scale scenes.

We first use DVGO, a popular accelerated NeRF variant, as the representative of the NeRF methods because training the accelerated NeRF variants is more environment-friendly and can significantly reduce the energy costs and carbon emissions of our work. We also evaluate vanilla NeRF and NeuS as two backbones on T & T. Because the vanilla NeRF is still a common baseline in related studies, while NeuS is a popular method which combines volume rendering and surface reconstruction.

The quantitative results in Table 1 and Table 2 support that NFC consistently improves three representative classes of existing neural field methods. We display the qualitative results in Figure 3. Particularly, DVGO, the accelerated variant which sometimes suffers from reconstructing challenging scenes, can be strongly enhanced, while vanilla NeRF, a relatively slow neural field method, can

Table 1: Quantitative results of DVGO on Replica Dataset.

| Scene | Mode | PSNR(↑) | SSIM(↑) | LPIPS(↓) |
|---|---|---|---|---|
| Scene 1 | Regression | 13.03 | 0.508 | 0.726 |
| | Classification | **34.63** | **0.934** | **0.0582** |
| Scene 2 | Regression | 14.81 | 0.654 | 0.640 |
| | Classification | **33.82** | **0.942** | **0.0660** |
| Scene 3 | Regression | 15.66 | 0.661 | 0.634 |
| | Classification | **34.04** | **0.965** | **0.0451** |
| Scene 4 | Regression | 18.17 | 0.696 | 0.546 |
| | Classification | **36.52** | **0.977** | **0.0311** |
| Scene 5 | Regression | 15.17 | 0.640 | 0.504 |
| | Classification | **35.93** | **0.974** | **0.0576** |
| Scene 6 | Regression | 21.33 | 0.854 | 0.254 |
| | Classification | **29.75** | **0.941** | **0.0994** |
| Scene 7 | Regression | 22.54 | 0.865 | 0.231 |
| | Classification | **34.77** | **0.966** | **0.0432** |
| Scene 8 | Regression | 15.89 | 0.724 | 0.519 |
| | Classification | **33.40** | **0.952** | **0.0775** |
| Mean | Regression | 17.08 | 0.700 | 0.507 |
| | Classification | **34.11** | **0.956** | **0.0598** |

Table 2: Quantitative results of DVGO, (vanilla) NeRF, NeuS on T&T Dataset. The mean metrics are computed over four scenes of T&T.

| Model | Mode | PSNR(↑) | SSIM(↑) | LPIPS(↓) |
|---|---|---|---|---|
| DVGO | Regression | 22.41 | 0.776 | 0.236 |
| | Classification | **23.18** | **0.810** | **0.178** |
| NeRF | Regression | 22.16 | 0.679 | 0.382 |
| | Classification | **22.68** | **0.716** | **0.315** |
| NeuS | Regression | 19.97 | 0.620 | 0.413 |
| | Classification | **21.67** | **0.679** | **0.317** |

Table 3: Quantitative results of D-NeRF on dynamic scenes, LEGO and Hook.

| Scene | Mode | PSNR(↑) | SSIM(↑) | LPIPS(↓) |
|---|---|---|---|---|
| Lego | Regression | 21.64 | 0.839 | 0.165 |
| | Classification | **23.11** | **0.886** | **0.121** |
| Hook | Regression | 29.25 | 0.965 | 0.118 |
| | Classification | **29.45** | **0.967** | **0.0392** |

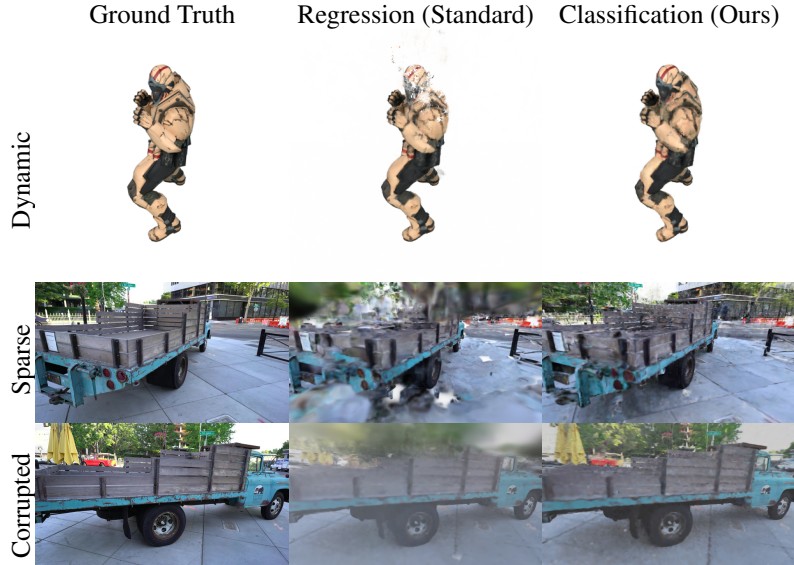

Figure 4: Qualitative comparison of NFC and NFR. Top Row: Dynamic Scenes. Middle Row: Sparse Inputs. Bottom Row: Corrupted Images.

also be consistently enhanced. The image quality improvement of DVGO on Replica Dataset is notably impressive with a 99.7% PSNR gain. Given the difficulties in reconstructing complex scenes in Replica, such as intricate texture patterns in complex objects, varying light condition and sparse training views, the advantages of NFC over NFR become more significant.

**Dynamic Scene** Dynamic scene rendering is a popular but more challenging task, which requires the capability of neural fields to model time domain. A NeRF variant, D-NeRF, has been specifically designed to render dynamic scenes. We also study how NFC improves D-NeRF on two common dynamic scenes, Lego and Hook (Pumarola et al., 2021). The quantitative results in Table 3 demonstrate that D-NeRF-C can render dynamic scenes better. The qualitative results of dynamic scenes in the top row of Figure 4 shows that that D-NeRF-C can produce better quality rendering results than D-NeRF-R. Both quantitative and qualitative results demonstrate that D-NeRF-C significantly surpasses D-NeRF-R.

**Challenging Scenes** NeRF imposes high requirements on data collection, including relatively dense training views, static illumination conditions, and precise camera calibration. These conditions can be particularly challenging due to inherent errors or difficulties in data collection.

While common scenes in NeRF studies often look clear, the training images in practice may be limited or corrupted due to realistic difficulties or inherent errors in data collection. For example, collected digital images often contain Gaussian noise due to the sensor-related factors Boyat and Joshi (2015). Robustness to data corruption and noise memorization can be an important performance metric for neural networks in weakly-supervised learning but rarely touched by previous NeRF studies.

To further understand the effectiveness and robustness of NFC, we empirically study NFC and NFR on two challenging tasks: **(1) novel view synthesis with sparse inputs** and **(2) novel view synthesis with image corruption**. Our benchmark for these two tasks is the Truck scene in the T & T Intermediate dataset, with DVGO selected as our backbone due to its adaptability.

Table 4: Quantitative results of neural rendering with sparse training images.

| Data Size | Mode | PSNR($\uparrow$) | SSIM($\uparrow$) | LPIPS($\downarrow$) |
|---|---|---|---|---|
| 20% | Regression | 14.87 | 0.530 | 0.580 |
| | Classification | **19.38** | **0.629** | **0.395** |
| 40% | Regression | 18.76 | 0.637 | 0.426 |
| | Classification | **21.73** | **0.711** | **0.340** |
| 60% | Regression | 21.02 | 0.682 | 0.394 |
| | Classification | **22.27** | **0.728** | **0.329** |
| 80% | Regression | 21.72 | 0.698 | 0.386 |
| | Classification | **22.46** | **0.734** | **0.322** |

Table 5: Quantitative results of neural rendering with corrupted training images.

| Noise Scale | Mode | PSNR($\uparrow$) | SSIM($\uparrow$) | LPIPS($\downarrow$) |
|---|---|---|---|---|
| 0.2 | Regression | 21.97 | 0.694 | 0.406 |
| | Classification | **22.33** | **0.719** | **0.353** |
| 0.4 | Regression | 21.33 | 0.663 | 0.451 |
| | Classification | **22.08** | **0.692** | **0.391** |
| 0.6 | Regression | 19.67 | 0.615 | 0.512 |
| | Classification | **21.45** | **0.662** | **0.429** |

First, in the sparse-input task, we train DVGO with the sparse version of a simple Truck scene, called Sparse Truck, where we randomly remove some training images. We visualize the qualitative comparisons of NFC and NFR with sparse inputs in the middle row of Figure 4. The experimental results in Table 4 and Figure 10 (of the appendix) further support that the advantage of NFC over NFR becomes even more significant with fewer training inputs.

Second, in the image-corruption task, we inject Gaussian noise with the standard deviation as $\mathrm{std}$ into original Truck images (each color value lies in $[0, 1]$) and obtain the corrupted version, called Corrupted Truck. We visualize the qualitative comparisons of NFC and NFR with corrupted images in Figure 4. The experimental results in Table 5 and Figure 11 (of the appendix) further support that the advantage of NFC over NFR is robust to image corruption.

NFC can encounter the challenging scenes significantly better than NFR with the difficulties in real-world data collection. This makes the proposed NFC even more competitive in the real-world practice. This observation aligns with the significant improve observed for the Replica Dataset, which is exactly a group of real-world challenging scenes.

## 4.2 Neural Surface Reconstruction Experiments

Surface reconstruction or geometry reconstruction is a fundamental task in both computer vision and computer graphics. Recent neural surface reconstruction methods Yariv et al. (2020; 2021); Oechsle et al. (2021); Wang et al. (2021) represent another class of neural field method. To evaluate the effectiveness of NFC for surface reconstruction, we choose a popular neural surface reconstruction method, NeuS, as the backbone which can reconstruct both RGB images and surface information. The training of NeuS requires no ground-truth surface information.

Given the requirement for both image and geometric assessments in surface reconstruction tasks, we employ the Replica Dataset, which provides comprehensive visual and geometric ground-truth for both image and geometry evaluation, and the T & T dataset for image quality evaluation. To assess geometric quality, we employ a set of popular geometry quality metrics, including Chamfer-$\ell_1$ Distance, F-score, when ground-truth surface and geometric information is available.

In terms of geometric assessment, our NeuS-C model achieves highly impressive improvements. The quantitative results in Table 6 demonstrate that all geometric metrics show impressive improvements

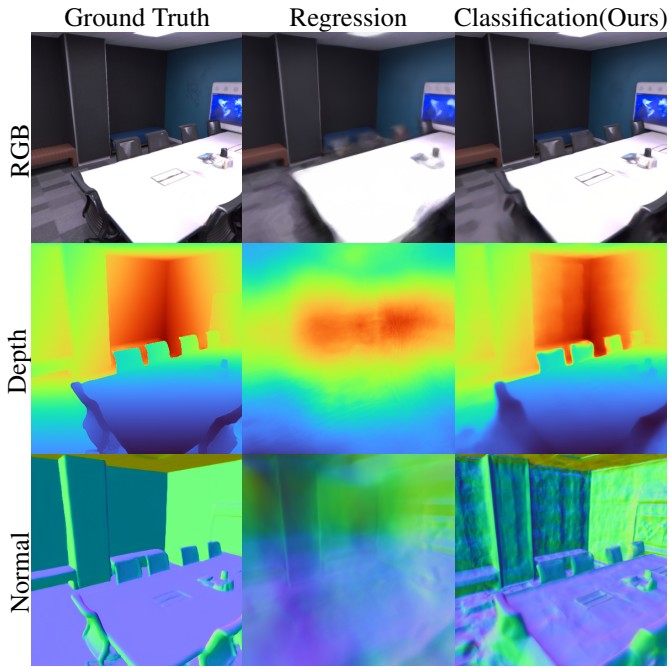

Figure 5: Qualitative comparisons of RGB rendering, depth rendering and normal rendering between NeuS-R and NeuS-C for neural surface reconstruction. Dataset: Replica Scene 8 (Office 4).

Table 6: Quantitative results of neural surface reconstruction on Replica. Model: NeuS.

| Scene | Training | PSNR(↑) | SSIM(↑) | LPIPS(↓) | Chamfer-$\ell_1$(↓) | Accuracy(↓) | Completeness(↓) | Precision(↑) | Recall(↑) | F-score(↑) | Normal C.(↑) |
|---|---|---|---|---|---|---|---|---|---|---|---|
| Scene 1 | Regression | 28.52 | 0.843 | 0.148 | 18.13 | 11.60 | 24.65 | 29.84 | 17.79 | 22.29 | 72.14 |
| | Classification | **31.12** | **0.848** | **0.101** | **6.879** | **6.841** | **6.917** | **59.42** | **59.98** | **59.70** | **77.59** |
| Scene 2 | Regression | 29.15 | 0.834 | 0.179 | 19.00 | 12.73 | 25.27 | 28.32 | 16.29 | 20.69 | 75.16 |
| | Classification | **31.27** | **0.844** | **0.148** | **8.582** | **8.380** | **8.785** | **48.88** | **47.24** | **48.05** | **75.86** |
| Scene 3 | Regression | 28.44 | 0.877 | 0.150 | 40.34 | 33.80 | 46.89 | 9.474 | 5.680 | 7.102 | 67.69 |
| | Classification | **33.99** | **0.925** | **0.0778** | **8.022** | **7.906** | **8.138** | **53.23** | **51.76** | **52.48** | **76.00** |
| Scene 4 | Regression | 31.84 | 0.874 | 0.152 | 18.90 | 13.61 | 24.19 | 22.25 | 12.52 | 16.02 | 71.89 |
| | Classification | **37.44** | **0.939** | **0.0572** | **5.162** | **5.051** | **5.272** | **68.01** | **72.97** | **70.40** | **78.59** |
| Scene 5 | Regression | 33.78 | 0.897 | 0.121 | 76.33 | 11.83 | 140.83 | 33.09 | 0.6030 | 1.184 | 58.05 |
| | Classification | **37.59** | **0.940** | **0.0588** | **11.53** | **11.46** | **11.60** | **38.54** | **35.44** | **36.93** | **70.26** |
| Scene 6 | Regression | 27.82 | 0.882 | 0.141 | 15.63 | 11.71 | 19.54 | 30.74 | 21.53 | 25.32 | 69.33 |
| | Classification | **31.30** | **0.902** | **0.102** | **8.916** | **8.348** | **9.484** | **52.21** | **48.61** | **50.35** | **73.01** |
| Scene 7 | Regression | 27.19 | 0.867 | 0.135 | 22.77 | 16.71 | 28.83 | 20.54 | 12.14 | 15.26 | 73.81 |
| | Classification | **31.28** | **0.915** | **0.0773** | **6.151** | **5.460** | **6.842** | **70.22** | **62.44** | **66.10** | **83.43** |
| Scene 8 | Regression | 28.29 | 0.908 | 0.130 | 30.82 | 25.71 | 35.92 | 15.34 | 8.730 | 11.13 | 68.86 |
| | Classification | **35.43** | **0.948** | **0.0564** | **6.263** | **5.910** | **6.615** | **60.93** | **59.94** | **60.43** | **80.54** |
| Mean (8 scenes) | Regression | 29.38 | 0.873 | 0.145 | 30.24 | 17.21 | 43.26 | 23.70 | 11.91 | 14.87 | 69.62 |
| | Classification | **33.68** | **0.908** | **0.0848** | **7.689** | **7.420** | **7.957** | **56.43** | **54.80** | **55.56** | **76.91** |

with NFC across all eight scenes. For example, over the eight surface reconstruction tasks, the mean Chamfer-$\ell_1$ Distance drops by $74.6\%$, while the mean F-score has improved by $273\%$. Most surface quality metrics have been improved by more than 10 points. In terms of image quality, NFC also exhibits substantial improvements in all image quality metrics over the eight scenes. The qualitative results in Figure 5 show that NFC significantly improves NeuS in both surface reconstruction and RGB rendering. These results quantitatively demonstrate the general effectiveness of NFC in improving neural fields for surface reconstruction.

### 4.3 DISCUSSION AND ABLATION STUDY

In this section, we further discuss the effectiveness and efficiency of NFC.

**Ablation Study** We conduct ablation study on Target Encoding and Classification Loss using DVGO and NeuS on Replica Dataset in Tables 7 and 8 , as well as Table 13 of the appendix. The quantitative results demonstrate that Target Encoding is usually helpful and lead to expected performance improvements, while the Classification Loss term plays a dominant role in the improvements of NFC.

Table 7: Ablation study I. Model: DVGO. Dataset: Replica Scene 6. GPU: A100.

| Mode | PSNR(↑) | SSIM(↑) | LPIPS(↓) | Training Time(minutes) | Rendering Time(seconds) |
|---|---|---|---|---|---|
| NFR (Standard) | 21.33 | 0.854 | 0.254 | 9.33 | 0.312 |
| NFC w/o Target Encoding | 27.99 | 0.935 | 0.110 | 9.76 | 0.312 |
| NFC (Ours) | **29.75** | **0.941** | **0.0994** | 9.93 | 0.328 |

Table 8: Ablation study II. Model: NeuS. Dataset: Replica Scene 7. GPU: A100.

| Mode | PSNR(↑) | SSIM(↑) | LPIPS(↓) | Training Time(hours) | Rendering Time(seconds) |
|---|---|---|---|---|---|
| NFR (Standard) | 27.19 | 0.867 | 0.135 | 2.84 | 10.27 |
| NFC w/o Target Encoding | 30.03 | 0.899 | 0.0996 | 2.92 | 10.27 |
| NFC (Ours) | **31.28** | **0.915** | **0.0773** | 2.98 | 10.83 |

We believe that it may be fine enough to employ the classification loss technique alone in some cases, while two components are both useful.

**Computational Cost** We study the computational costs of NFC and NFR in Tables 7 and 8. It shows that the extra training cost of NFC is very limited (+6% for DVGO and +4% for NeuS) compared with the significant empirical improvement, and the extra rendering cost is nearly zero.

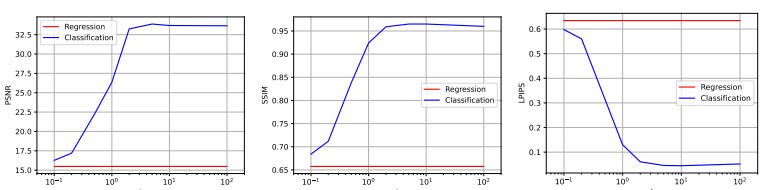

Figure 6: The curves of PSNR, SSIM, and LPIPS with respect to the hyperparameter $\lambda$. NFC is robust to a wide range of $\lambda$. Model: DVGO. Dataset: Replica Scene 3.

Figure 7: The learning curves of NFC and NFR.

**Robustness to the hyperparameter** $\lambda$ The hyperparameter $\lambda$ controls the weight of the classification loss. We plot how the value of $\lambda$ affects the model performance in Figure 6. The quantitative results shows that a wide value range of $\lambda$ can enhance the performance. This suggests that the proposed NFC can be easily employed in practice with the default hyperparameter and limited fine-tuning cost.

**Generalization** We visualize the training and test PSNR curves of NFR and NFC using DVGO on Replica Scene 6 in Figure 7. We observed that NFC has a slightly better training PSNR and a much better test PSNR. This suggests that the main gain of NFC comes from generalization rather than optimization. It is known that generalization closely relate to flatter minima of loss landscape (Hochreiter and Schmidhuber, 1997; Xie et al., 2021b), while neural fields research largely lacks generalization analysis. We leave theoretical interpretations as future work.

**Limitations** The main limitation is that NFC can often remarkably improve generalization, only when poor generalization indeed exist. However, when tasks are simple or neural fields have strong view generalization, the improvements become limited or even negligible (see appendix).

## 5 CONCLUSION

In this work, we visited a very fundamental but overlooked topic for neural field methods: regression versus classification. Then we design a NFC framework which can formulate existing neural field methods as classification models rather than regression models. Our extensive experiments support that Target Encoding and classification loss can significantly improve the performance of most existing neural field methods in novel view synthesis and geometry reconstruction. Moreover, the improvement of NFC is robust to sparse inputs, image noise, and dynamic scenes. While our work mainly focuses on 3D vision and reconstruction, we believe NRC is a general neural field framework. We believe it will be very promising to explore and enhance the generalization of neural fields.

ACKNOWLEDGMENT

This work was partly supported by the National Natural Science Foundation of China under Grant No. 31771475.

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

## A  EXPERIMENTAL SETTINGS AND DETAILS

In this section, we present the experimental settings and details for reproducing the results. The main principle of our experimental setting is to fairly compare NFC and NFR for NeRF and the variants. Our experimental settings follows original papers to produce the baselines, unless we specify them otherwise.

**Classification Loss Setting** We fine-tune the classification loss weight $\lambda$ from $\{1, 2, 5, 10\}$ for the NeRF family and $\{1, 2, 5, 10, 20, 50, 100\}$ for the NeuS family.

### A.1  MODELS AND OPTIMIZATION

**DVGO Setting** We employ the sourcecode of DVGO (Version 2) in the original project (Sun et al., 2022) without modifying training hyperparameters. So we train DVGO via Adam (Kingma and Ba, 2015) with the batch size $B = 4096$. The learning rate of density voxel grids and color/feature voxel grids is 0.1, and the learning rate of the RGB net (MLP) is 0.001. The total number of iterations is 30000. We multiply the learning rate by 0.1 per 1000 iterations.

**NeRF Setting** We employ a popular open-source implementation (Yen-Chen, 2020) of the original NeRF. Again, we follow its defaulted training setting. The learning rate is 0.0005, and the learning rate scheduler is $0.1^{iters/500000}$.

**D-NeRF Setting** We directly employ the sourcecode of D-NeRF in the original project (Pumarola et al., 2021). The learning rate is 0.0005. The total number of iterations is 800k. The learning rate decay follows the original paper.

**NeuS Setting** We employ the NeuS implementation of SDFStudio (Yu et al., 2022) and follow its default hyperparameters. The difference of the hyperparameters between SDFStudio and the original paper (Wang et al., 2021) is that SDFStudio trains 100k iterations, while the original paper trains 300k iterations.

**Strivec Setting** We directly employ the sourcecode of Strivec in the original project (Gao et al., 2023) and follow its original training setting on the classical Chair scene.

**4DGS Setting** We directly employ the sourcecode of 4DGS (Wu et al., 2023) and follow its original training setting on the classical Mutant scene. As the target encoding is relatively expensive for Gaussian Splatting, we use NFC-4DGS without target encoding as our method.

### A.2  DATASETS

**Replica Dataset** Replica Dataset has no splitted training dataset and test dataset. In the experiments on Replica, if one image index is divisible by 10, we move the image to the test dataset; if not, we move the image to the training dataset. Thus, we have $90\%$ images for training and $10\%$ images for evaluation.

**T&T Dataset Advanced** T&T Dataset Advanced has no splitted training data and test data. We follow the original splitted way in the standard setting. In the experiments on T&T Dataset Advanced, if one image index is divisible by 10, we move the image to the test T&T Dataset Advanced; if not, we move the image to the training dataset. Similarly, we again have $90\%$ images for training and $10\%$ images for evaluation.

**T&T Dataset Intermediate** T&T Dataset Intermediate has splitted training data and test data. We follow the original splitted way in the standard setting. In the experiments of sparse inputs, we randomly remove the training images. In the experiments of corrupted images, we inject Gaussian noise with the scale std into color values of the training images, and clip the corrupted color values into $[0, 1]$.

**Dynamic Scenes: LEGO, Hook, and Mutant** The three dynamics scenes are used in the original D-NeRF paper. We use them in the same way without any modification.

**Chair Scene from NeRF Synthetic** We also use the classical Chair scene for evaluating NFC and NFR with Strivec (Gao et al., 2023). Since the standard Chair scene is a quite simple benchmark, we only use $20\%$ training images of Chair to obtain a more challenging benchmark in our experiment.

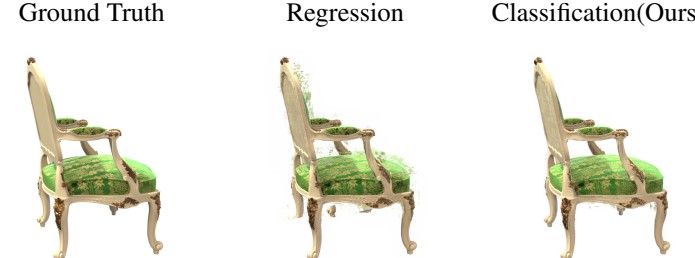

Ground Truth      Regression      Classification(Ours)

Figure 8: Qualitative comparisons of Strivec-C and Strivec-R for the classical Chair scene.

## B  SUPPLEMENTARY EMPIRICAL ANALYSIS AND DISCUSSION

In this section, we present supplementary experimental results.

We present the quantitative results of DVGO, NeRF, and NeuS on each T&T scenes in Tables 9, 10, and 11, respectively.

We present the quantitative results and qualitative results of Trivec over the Chair scene in Table 12 and Figure 8, respectively. We present the qualitative results of 4DGS (Wu et al., 2023) in Figure 9. NFC also improves the performance of Trivec and 4DGS but *not that significantly*. Perhaps, it is because Trivec and 4DGS can generalize well enough for common scenes. We conjecture that the improvement of NFC is relatively significant for those challenging settings.

Table 9: Quantitative results of DVGO methods on T&T.

| Scene | Mode | PSNR(↑) | SSIM(↑) | LPIPS(↓) |
|---|---|---|---|---|
| Scene 1 | Regression | 21.80 | 0.759 | 0.243 |
|  | Classification | **22.04** | **0.787** | **0.192** |
| Scene 2 | Regression | 23.96 | 0.847 | 0.250 |
|  | Classification | **25.32** | **0.875** | **0.178** |
| Scene 3 | Regression | 18.64 | 0.697 | 0.272 |
|  | Classification | **19.93** | **0.758** | **0.197** |
| Scene 4 | Regression | 25.27 | 0.800 | 0.179 |
|  | Classification | **25.43** | **0.820** | **0.146** |
| Mean | Regression | 22.41 | 0.776 | 0.236 |
|  | Classification | **23.18** | **0.810** | **0.178** |

**Sparse inputs** We plot the curves of PSNR, SSIM, and LPIPS with respect to the training data size to compare NFC and NFR in Figure 10.

**Robustness to corrupted images** We plot the curves of PSNR, SSIM, and LPIPS with respect to the image noise scale std to compare NFC and NFR in Figure 11.

Table 10: Quantitative results of (vanilla) NeRF on T&T.

| Scene | Mode | PSNR(↑) | SSIM(↑) | LPIPS(↓) |
|---|---|---|---|---|
| Scene 1 | Regression | 22.20 | 0.682 | 0.349 |
|  | Classification | **22.24** | **0.691** | **0.342** |
| Scene 2 | Regression | 23.97 | 0.799 | 0.374 |
|  | Classification | **25.01** | **0.835** | **0.289** |
| Scene 3 | Regression | 18.63 | 0.548 | 0.513 |
|  | Classification | **19.36** | **0.628** | **0.369** |
| Scene 4 | Regression | 23.87 | 0.688 | 0.293 |
|  | Classification | **24.12** | **0.713** | **0.262** |
| Mean | Regression | 22.17 | 0.679 | 0.382 |
|  | Classification | **22.68** | **0.717** | **0.315** |

Table 11: Quantitative results of NeuS on T&T.

| Scene | Mode | PSNR(↑) | SSIM(↑) | LPIPS(↓) |
|---|---|---|---|---|
| Scene 1 | Regression | 20.73 | 0.636 | 0.393 |
| | Classification | **21.31** | **0.682** | **0.310** |
| Scene 2 | Regression | 21.26 | 0.739 | 0.434 |
| | Classification | **22.70** | **0.794** | **0.290** |
| Scene 3 | Regression | 17.58 | 0.551 | 0.428 |
| | Classification | **19.08** | **0.601** | **0.345** |
| Scene 4 | Regression | 20.32 | 0.554 | 0.398 |
| | Classification | **22.59** | **0.640** | **0.322** |
| Mean | Regression | 19.97 | 0.620 | 0.413 |
| | Classification | **21.67** | **0.679** | **0.317** |

Table 12: Quantitative results of Strivec on the Chair scene.

| Scene | Mode | PSNR(↑) | SSIM(↑) | LPIPS(↓) | Training Time(hours) | Rendering Time(seconds) |
|---|---|---|---|---|---|---|
| Chair | Regression | 28.53 | 0.944 | 0.0672 | 1.16 | 4.38 |
| | Classification | **32.10** | **0.969** | **0.0217** | 1.23 | 4.59 |

**Ablation study over more scenes** We present the ablation study of DVGO over all 8 Replica scenes in Tables 13. The results show that the classification loss plays a dominant role in the proposed NFC, while the Target Encoding module is still often helpful is most scenes (75%) and lead to expected performance improvements.

**Generalization** We are surprised by the observation in Figure 7 that the generalization gap becomes the main challenge of neural fields in these experiments. Even if neural field models can fit training data very well, they may fail unexpectedly sometimes. According to existing theoretical knowledge in deep learning, this indicates that loss landscape of neural field models has a lot of sharp minima that generalize poorly. This pitfall is possibly mitigated by multiple ways that help selecting flat minima. While this work focuses on reshaping loss landscape, our previous theoretical studies (Xie et al., 2021c; 2022b; 2023a) also explore how to find flatter minima with advanced training algorithms only. They are designed for general neural networks and likely also work well for neural fields. We tend to believe these studies may also shed lights on future directions of theoretically understanding and principally improving (generalization of) neural fields in a plug-and-play way without modifying the model architectures.

Ground Truth         Regression (Standard)         Classification (Ours)

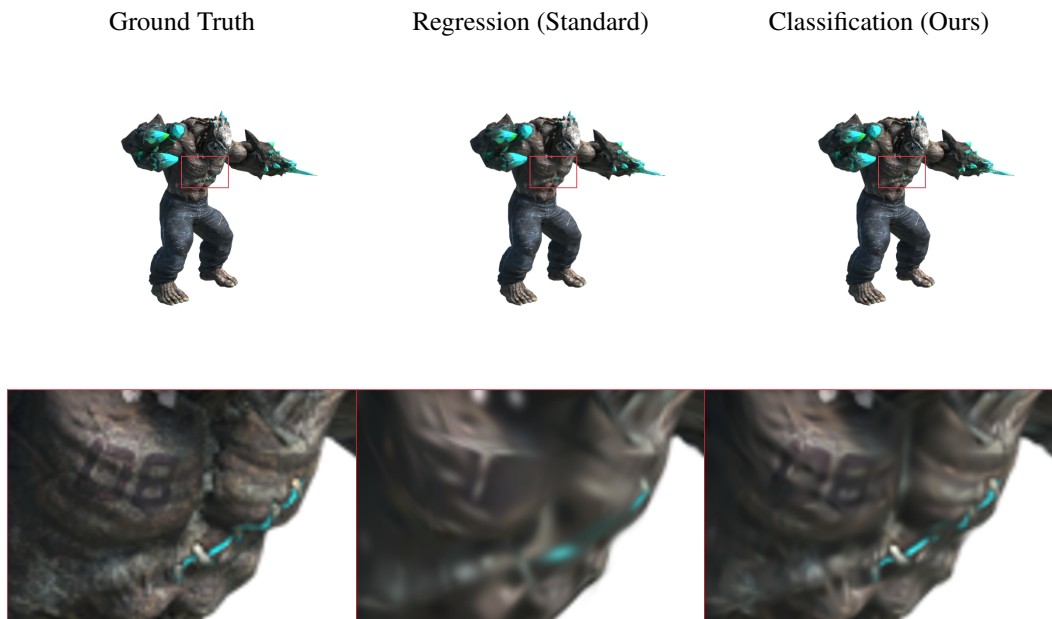

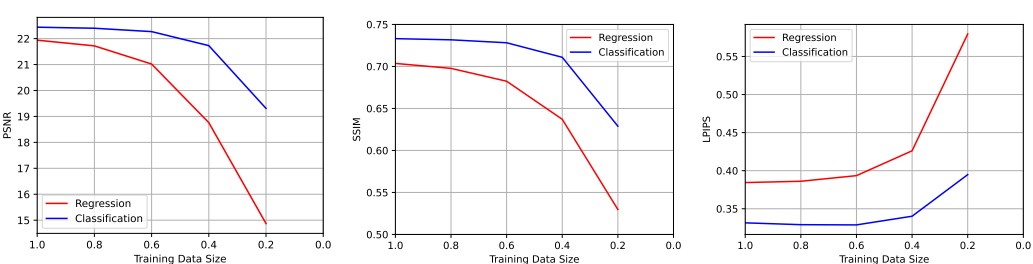

Figure 9: Qualitative comparison of 4DGS-C and 4DGS-R for the dynamic Mutant scene. LIPIS: $0.0167 \rightarrow 0.009$.

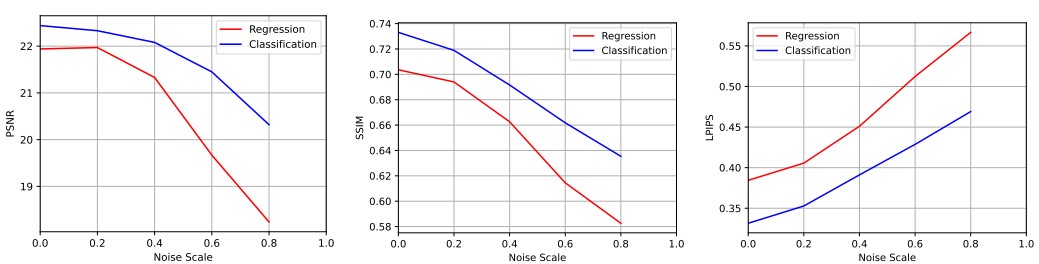

Figure 10: We plot the curves of PSNR, SSIM, and LPIPS with respect to the training data size, namely the portion of training samples kept from the original training dataset. The improvement of NFC is even more significant when the training data size decreases. Model: DVGO. Dataset: T&T-Truck.

Figure 11: We plot the curves of PSNR, SSIM, and LPIPS with respect to the image noise scale $\text{std}$. The improvement of NFC is more significant when training image is corrupted by random noise. Model: DVGO. Dataset: T&T-Truck.

Table 13: Ablation study. Model: DVGO. Dataset: Replica 8 scenes.

| Scene | Mode | PSNR($\uparrow$) | SSIM($\uparrow$) | LPIPS($\downarrow$) |
|---|---|---|---|---|
| Scene 1 | NFR (Standard) | 13.03 | 0.508 | 0.726 |
|  | NFC w/o Target Encoding | 32.52 | 0.929 | 0.0718 |
|  | NFC (Ours) | **34.63** | **0.934** | **0.0582** |
| Scene 2 | NFR (Standard) | 14.81 | 0.654 | 0.640 |
|  | NFC w/o Target Encoding | **33.92** | **0.951** | **0.0615** |
|  | NFC (Ours) | 33.82 | 0.942 | 0.0660 |
| Scene 3 | NFR (Standard) | 15.66 | 0.661 | 0.634 |
|  | NFC w/o Target Encoding | 28.69 | 0.954 | 0.0715 |
|  | NFC (Ours) | **34.04** | **0.965** | **0.0451** |
| Scene 4 | NFR (Standard) | 18.17 | 0.696 | 0.546 |
|  | NFC w/o Target Encoding | **37.35** | 0.976 | 0.0329 |
|  | NFC (Ours) | 36.52 | **0.977** | **0.0311** |
| Scene 5 | NFR (Standard) | 15.17 | 0.640 | 0.504 |
|  | NFC w/o Target Encoding | 35.86 | 0.970 | 0.0595 |
|  | NFC (Ours) | **35.93** | **0.974** | **0.0576** |
| Scene 6 | NFR (Standard) | 21.33 | 0.854 | 0.254 |
|  | NFC w/o Target Encoding | 27.99 | 0.935 | 0.110 |
|  | NFC (Ours) | **29.75** | **0.941** | **0.0994** |
| Scene 7 | NFR (Standard) | 22.54 | 0.865 | 0.231 |
|  | NFC w/o Target Encoding | 27.66 | 0.903 | 0.153 |
|  | NFC (Ours) | **34.77** | **0.966** | **0.0432** |
| Scene 8 | NFR (Standard) | 15.89 | 0.724 | 0.519 |
|  | NFC w/o Target Encoding | **36.11** | **0.967** | **0.0638** |
|  | NFC (Ours) | 33.40 | 0.952 | 0.0775 |
| Mean | NFR (Standard) | 17.08 | 0.700 | 0.507 |
|  | NFC w/o Target Encoding | 32.49 | 0.949 | 0.0676 |
|  | NFC (Ours) | **34.11** | **0.956** | **0.0598** |

