# OpenReview forum: "Neural Field Classifiers via Target Encoding and Classification Loss"
_ICLR.cc/2024/Conference — ICLR 2024 poster_

### Official Review · Reviewer_dKd6 · 2023-10-27

**Soundness:** 3 good
**Presentation:** 3 good
**Contribution:** 3 good
**Rating:** 8
**Confidence:** 3

**Summary:**

This paper proposes to formulate the neural field methods as a classification problem rather than the traditional regression problem. To achieve this, the continuous RGB value is encoded as an 8-bit binary (discrete) vector, and BCE loss is used to train the model. Experiments on novel view synthesis and neural surface reconstruction validate that the proposed classification method is superior to its regression variant.

**Strengths:**

1. The motivation is clear, and the readability is satisfying to me (not an expert in Nerf);

2. The proposed method is simple yet effective;

3. Experiments on various datasets are conducted to validate the effectiveness of the method.

**Weaknesses:**

1. The ablation study is insufficient:
    - It can be observed from Eq. (6) that the original regression loss is still included. Although the authors pointed out that the BCE loss is significantly larger than the regression loss, I'm still curious about the effect of using BCE loss only. Since the BCE loss is more dominant, is the regression loss essential?
    - It seems that without text encoding, the performance is close to the final version. Moreover, only two scenes of one dataset are used to conduct the ablation study. I'm wondering, on average (all scenes and other datasets), is the text encoding essential, or is it optimal?

2. The implementation and comparison are conducted based on methods published in 2020 and 2021. Only DVGO was published in CVPR 2022, which was also long ago. It is kindly suggested to conduct experiments on more recent SOTA methods to evaluate the effectiveness and generality of the proposed approach.

**Questions:**

See weaknesses.

Note: Since I'm not an expert in Nerf, I'm truly wondering, why such a simple yet effective formulation is proposed in 2023.

---

> ### Author Response · Authors · 2023-11-20
> **Responses to Reviewer dKd6**
>
> We sincerely appreciate Reviewer dKd6’s kind support and helpful comments.
>
> We properly addressed your concerns below.
>
> Q1: I'm still curious about the effect of using BCE loss only. Since the BCE loss is more dominant, is the regression loss essential?
>
> A1: A good question. Although the (slightly modified with ) BCE is more dominant, the regression loss is still helpful. We note that the the modification (the clip) of the classification loss is desired for the numerical stability as the integrated probability through volume rendering may be slightly greater than or equal to one. However, the clip operation is non-differentialable for some data samples where the predicted target $\hat{C}\geq 1-\epsilon$. For these samples, the classification loss can only produce zero gradient. This explains when the regression loss dominate the classification loss and why the regression loss is still usually helpful.
>
> Q2: It seems that without text encoding, the performance is close to the final version. Moreover, only two scenes of one dataset are used to conduct the ablation study. I'm wondering, on average (all scenes and other datasets), is the text encoding essential, or is it optimal?
>
> A2: We respectfully note that the PSNR gains in our two ablation studies are 1.76 and 1.25, and such PNSR gains can be considered to be very significant in NeRF studies. Most new NeRF methods cannot outperform the baselines by more than 1 PSNR gain. If the reviewer want to know more about recent empirical advancements of NeRF studies, please refer to recent top-conference NeRF papers ( e.g. Trivec ICCV2023).
>
> Thus, while the classification loss itself can lead to a more significant improvement, we hold that the target encoding is still expected to be helpful in most cases.
>
> In fact, the regression loss (discussed in Q1-A1) and target encoding are both helpful and positive, while their effects are less significant than the (modified) BCE loss term.
>
> Q3: The implementation and comparison are conducted based on methods published in 2020 and 2021. Only DVGO was published in CVPR 2022, which was also long ago. It is kindly suggested to conduct experiments on more recent SOTA methods to evaluate the effectiveness and generality of the proposed approach.
>
> A3: Thanks for the constructive suggestion. The currently used NeRF backnones are all popular and representative methods covering the tasks of static scenes, geometry reconstruction, and dynamic scenes. The experiments are comprehensive.
>
> Some recent works proposed in 2023 (e.g. Strivec) are considered to be SOTA. In our recent supplementary experiment, Strivec-C can outperform Strivec-R by 3 PSNR gain. We are conducting more comparative experiments using SOTA methods proposed in 2023. We would like to present more empirical results of recent NeRF variants for comparing NFC and NFR in the revision.
>
> Q4: Since I'm not an expert in Nerf, I'm truly wondering, why such a simple yet effective formulation is proposed in 2023.
>
> A4: An interesting question. We personally tend to think that most researchers from the NeRF community focus more on modifying backbones and lack the motivation/tradition to rethink the basic formula of NeRF from a learning perspective. Moreover, the numerical instability problem of directly using the BCE loss may have prevented some researchers’ early explorations along this direction.
>
>
> Reference:
>
> [1] Gao, Q., Xu, Q., Su, H., Neumann, U., & Xu, Z. (2023). Strivec: Sparse tri-vector radiance fields. In Proceedings of the IEEE/CVF International Conference on Computer Vision (pp. 17569-17579).

---

> > ### Comment · Reviewer_dKd6 · 2023-11-21
> >
> > Thanks for the explanation, which addresses most of my concerns. I'm still curious about the ablation study of text encoding. On average (other datasets and scenes), how does it improve the performance? Moreover, I'm wondering if there are other design choices for text encoding, and how they perform.

---

> > > ### Author Response · Authors · 2023-11-21
> > > **About the ablation study of Target Encoding.**
> > >
> > > Q5: On average (other datasets and scenes), how does target encoding improve the performance?
> > >
> > > A5: We would like to extend the ablation study of target encoding over more scenes. In the recent ablation study over 8 scenes, we observe that the PSNR gain of target encoding is 1.62, which is also significant.
> > >
> > > Q6: If there are other design choices for text encoding, and how they perform.
> > >
> > > A6: Thanks for the question. In principle, any inverse encoding algorithm is a possible choice for target encoding in neural fields. We have tried a very naïve strategy which formulates a 256-class classification problem via 256-dimensional one-hot encoding for each channel. This is unfortunately more expensive and weaker than the binary-number encoding. The employed binary-number encoding are very simple yet efficient as an inverse encoding method.
> > >
> > > We believe it will be interesting to explore learnable target encoding via a ML module in future. A learnable design choice could lead to further improvement beyond the rule-based design choice of this work.

---

> > > > ### Comment · Reviewer_dKd6 · 2023-11-21
> > > >
> > > > Thanks. I would like to raise my rating if the mentioned results are updated in the revision.

---

> ### Author Response · Authors · 2023-11-22
> **Paper Revision**
>
> We gratefully appreciate your kind support!
>
> We have revised our submission following your suggestions.
>
>
> -SOTA methods. We presented the quantitative and qualitative results of Strivec (ICCV 2023) in Table 12 and Figure 8, respectively. The results show that NFC may also improve very recent regression-based neural field methods. We will try to present more results of recent SOTA methods in future.
>
> -Ablation study. We presented the ablation study results over 8 scenes in Table 13. The results show that the classification loss plays a dominant role in the proposed NFC, while the Target Encoding module is still often helpful is most scenes (75%) and lead to expected performance improvements. We also revised our discussion on ablation study and note the dominant role of the classification in Section 4.3 (red-colored parts).

---

> > ### Comment · Reviewer_dKd6 · 2023-11-23
> >
> > After reading the responses and paper revision, I have raised my rating.

---

### Official Review · Reviewer_fXmy · 2023-11-01

**Soundness:** 3 good
**Presentation:** 3 good
**Contribution:** 3 good
**Rating:** 6
**Confidence:** 3

**Summary:**

In this paper, the authors proposed Neural Field Classifier that formulates existing neural field methods as classification tasks with target encoding and a classification loss. The authors conducted extensive experiments and visualizations to show that it is better than the regression based methods.

**Strengths:**

1. The formulation for classification with target encoding and BCE loss makes sense and is technically sound to me.
2. Extensive quantitative and qualitative showed that it is better than standard regression based methods.
3. Writing is good and easy to follow. The visualizations are also very informative and showed the improvement very clearly.

**Weaknesses:**

1. Besides the experiments and visualization, there is no theoretical analysis of why the proposed classification based formulation is better than the standard regression methods. It is because that the classification loss is easier to optimize or?
2. There is also little analysis on why it is more robust that the regression based methods?

**Questions:**

1. Why the proposed classification based formulation perform better than the standard regression based methods?
2. Why is more robust that regression based methods?

---

> ### Author Response · Authors · 2023-11-20
> **Responses to Reviewer fXmy**
>
> We highly appreciate Reviewer fXmy’s kind support and hard work.
>
> We properly addressed your concerns as follows.
>
> Q1: Besides the experiments and visualization, there is no theoretical analysis of why the proposed classification based formulation is better than the standard regression methods. It is because that the classification loss is easier to optimize or?
>
> A1: We frankly admit that there is no theoretical analysis yet. In fact, nearly all NeRF methods lack formal theoretical analysis about the rendering quality.
>
> However, according to the results in Figure 7, we may conjecture that the main advantage of NFC comes from better generalization of learned neural field models, while we also observe that optimizing the classification loss also leads to slightly better training quality. It will be a very promising future direction to explore why the classification loss may lead to better generalization.
>
> Q2: There is also little analysis on why it is more robust that the regression based methods?
>
> A2: Thanks for pointing out this promising research challenge. Similarly to Q1-A1, indeed, our work and related neural field studies did not formulate theoretical understanding of NeRFs. In fact, most 3D vision-related works lack formal theoretical understanding due to the methodology complexity.
>
> According to the intuition and existing empirical analysis, the advantages of NFC may come from multiple sources, including better training quality and better generalization. However, the formal theoretical mechanism is beyond the main scope of this work. We appreciate your suggestion and will explore the theoretical mechanism of generalization and robustness of neural field models in future.
>
> Finally, we sincerely thank the reviewer again for your kind support!

---

> > ### Comment · Reviewer_fXmy · 2023-11-21
> >
> > Thanks for the explanation. I do not have more questions and will keep my original positive rating.

---

### Official Review · Reviewer_WcH4 · 2023-11-01

**Soundness:** 3 good
**Presentation:** 3 good
**Contribution:** 3 good
**Rating:** 6
**Confidence:** 4

**Summary:**

The paper investigates whether regression can be replaced by classification for neural fields, particularly for the application of novel view synthesis. They propose a framework for transforming regression into classification using a binary encoding. They show that this formulation can effectively be optimized and perform experiments for novel view synthesis, the primary application of neural fields. They benchmark an array of NeRF methods and show that using classification loss improves novel view synthesis metrics by a significant margin. Also they show that classification leads to better robustness to corruption and better sample efficiency.

**Strengths:**

- The idea is to my knowledge the first to investigate classification as an objective for neural fields. The idea is interesting and applicable to most neural field architectures and is relatively agnostic to data type.

- The idea is general and could be applied to many different applications. I think this would be of interest to people in the neural field community.

- The results are strong and the experiments are thorough. The paper evaluates on multiple novel view synthesis datasets and benchmarks an array of NeRF variants.

The figures are illustrative and convey the main ideas of the paper.

**Weaknesses:**

Overall the weaknesses are relatively minor.

- The experiments are restricted to the domain of novel view synthesis.

- The qualitative results for the baselines look unreasonably poor. I would think that a vanilla NeRF model would look better on the scenes presented in the figures.

- Some parts of the method section are a little unclear. See questions.

**Questions:**

- How many training views are given for the Replica scenes? I couldn't find that detail in the paper. The results for regression in figure 3 look particularly poor. I would expect a crisper result for regression with a reasonable number of views.

- How is sigma being output by the MLP? Is that still being regressed, and if so why not encode it as a classification problem?

- How is the volume rendering part performed? Do you use the standard volume rendering equation and integrate the bits with sigma?

---

> ### Author Response · Authors · 2023-11-20
> **Responses to Reviewer WcH4**
>
> We sincerely appreciate Reviewer WcH4’s kind support and constructive comments.
>
> We addressed your concerns below.
>
> Q1: The experiments are restricted to the domain of novel view synthesis.
>
> A1: We admit that our experiments mainly focus on novel view synthesis, because the proposed classification paradigm is mainly designed for neural field methods. We also respectfully note that the experiments on geometry reconstruction in Tables 6 and 8 also support the advantage of our NFC method over the standard NFR method.
>
> Q2: The qualitative results for the baselines look unreasonably poor. I would think that a vanilla NeRF model would look better on the scenes presented in the figures.
>
> A2: Thanks for the suggestion. We presented the qualitative results of vanilla NeRF in Figure 3 Top Row. The regression baseline look well, while the classification variant is still significantly better. We would like to present more qualitative results of vanilla NeRF in the revision.
>
> Q3: How many training views are given for the Replica scenes? I couldn't find that detail in the paper. The results for regression in figure 3 look particularly poor. I would expect a crisper result for regression with a reasonable number of views.
>
> A3: We respectfully note that the details of dataset preprocessing including datset split of Replica are presented in Appendix A.2. We used 90% images (~90 views) for training and 10% images (~10 views) for evaluation.
>
> The results of NeuS-R on Replica in Table 6 actually look reasonably well, while the results of DVGO-R on Replica in Table 3 are indeed poor. As the training dataset are totally same, our results suggest that the accelerated NeRF variants like DVGO are not powerful enough complex scenes (e.g. Replica), which contain many details.
>
> However, for both NeuS and DVGO, our classifier-based method can significantly improve the rendering quality. Thus, we believe our experiments using DVGO and NeuS together can support the advantage of our NFC over the conventional NFR.
>
> Q4: How is sigma being output by the MLP? Is that still being regressed, and if so why not encode it as a classification problem?
>
> A4: A very good question. While, in principle, the density $\sigma$ can be encoded as a classification problem, we believe regressing $\sigma$ is a better choice for three reasons. First, most importantly, encoding $\sigma$ does not significantly improve the rendering quality in our early experiments. Second, the extra computational cost of encoding $\sigma$ is significantly larger than only encoding rgb, because, for example, an 8-channel $\sigma$ encoding may reqiure $8$ decoding costs for making once prediction. Thrid, some NeRF variants use two different network backbones for predicting the rgb color and the density, where encoding the density neural network may require more coding costs. In summary, the pitfalls (more costs and complexity) significantly outweigh its benefits (nearly zero performance improvement) .
>
> Q5: How is the volume rendering part performed? Do you use the standard volume rendering equation and integrate the bits with sigma?
>
> A5: We follow the standard volume rendering process. The only difference is that we integrate the predicted probabilities of each bits with the density.
>
>
> Reviewer WcH4 definitely recognizes the novelty and significance of our work.
>
> We would gratefully appreciate it, if the reviewer could insist on the opinion during the discussion and try to avoid a possible loss to the community.

---

> > ### Comment · Reviewer_WcH4 · 2023-11-23
> > **Reviewer Response**
> >
> > I thank the authors for their thorough response to my comments and questions. After reading other reviewer concerns and author comments I am maintaining my score at 6. I think the paper presents a general and interesting approach to neural fields, and could be of significant interest to the community.

---

### Official Review · Reviewer_zn3A · 2023-11-01

**Soundness:** 3 good
**Presentation:** 4 excellent
**Contribution:** 3 good
**Rating:** 6
**Confidence:** 4

**Summary:**

The paper proposes a novel Neural Field Classifier (NFC) framework that formulates existing neural field methods as classification tasks rather than regression tasks. The method also uses Target Encoding to encode continuous targets for regression tasks into discrete targets for classification tasks. The overall writing is clear. The experiments cover the comparison between the Regression model and the Classification model.


Raise score to 6 after rebuttal.

**Strengths:**

1. The overall writing is clear.
2. The experiments clearly show the improvement of the Classification model over the Regression model.

**Weaknesses:**

1. Some important reference is missing, such as [a].
2. [a] uses the CNN encoders to process the images. The idea of the proposed classifier looks similar to [a].
3. The biggest problem is the compared method. Among all the experiments, DVGO is compared extensively. However, DVGO is a method designed for acceleration, not for accuracy. It is unfair to compare the accuracy with a method not targeting accuracy.
4. What is the State-of-the-art accuracy of NeRF? Please compare with these methods regarding accuracy.
5. What is the computational cost of the proposed classifier-based methods? Please compare with DVGO regarding FLOPs and wall-clock time.


[a] pixelNeRF: Neural Radiance Fields from One or Few Images, CVPR 2021

**Questions:**

see weakness

---

> ### Author Response · Authors · 2023-11-20
> **Responses to Reviewer zn3A**
>
> We sincerely appreciate Reviewer zn3A’s hard work and constructive comments.
>
> We addressed your main concerns below.
>
> Q1: Some important reference is missing, such as [a]PixelNeRF.
>
> A1: Thanks for providing the reference. In fact, any existing regression-based NeRF variant could be the baseline. We would like to discuss more existing NeRF methods, including PixelNeRF.
>
> Q2: PixelNeRF uses the CNN encoders to process the images. The idea of the proposed classifier looks similar to PixelNeRF.
>
> A2: We are afraid that the reviewer may misunderstand our work or PixelNeRF. We respectfully argue that PixelNeRF and the proposed NF classification paradigm are two totally different methods. Our work does not use any CNN encoder to process the images. Our two contributions, Target Encoding and Classification Loss, are not touched by PixelNeRF. Moreover, the classification paradigm can be incorporated into the (regression-based) PixelNeRF rather than replacing.
>
> Q3: The biggest problem is the compared method. Among all the experiments, DVGO is compared extensively. However, DVGO is a method designed for acceleration, not for accuracy. It is unfair to compare the accuracy with a method not targeting accuracy.
>
> A3: Our work actually has only one baseline method, the regression paradigm compared with our classification paradigm. We kindly argue that, besides DVGO, we also train vanilla NeRF, NeuS, and D-NeRF as the backbone methods on 14 different scenes, including static scenes, geometry reconstruction, and dynamic scenes. The experiments are still comprehensive. Particularly, when the regression-based neural field models seriously overfit training views (as Figure 7 suggests), our classification-based method can significantly improve generalization and mitigate overfitting.
>
>
> Our prior backbone is the accelerated NeRF variant also because the accelerated variant is more environment-friendly and can significantly reduce the energy costs and carbon emissions of our work. In the line of NeRF research, people actually focused more on improving the rendering quality of the accelerated NeRF variants, because the efficiency rather than the quality (for standard static scenes) is often considered the main limitation of vanilla NeRF.
>
> Recent NeRF works, including Strivec (ICCV2023) and Zip-NeRF (ICCV2023), almost all aimed at (1) accurate and fast rendering, or (2) accurate rendering under some constrains (e.g. sparse/dynamic/corruption). Our experiments have covered these experimental settings.
>
>
> Q4: What is the State-of-the-art accuracy of NeRF? Please compare with these methods regarding accuracy.
>
> A4: Thanks for the helpful suggestion. Some recent works (e.g. Strivec ) proposed in 2023 are considered to be SOTA. In our recent supplementary experiment, Strivec-C can outperform Strivec-R by 3 PSNR gain. We are conducting more comparative experiments using SOTA methods proposed in 2023. We would like to present more empirical results of recent NeRF variants for comparing NFC and NFR in the revision.
>
> Q5: What is the computational cost of the proposed classifier-based methods? Please compare with DVGO regarding FLOPs and wall-clock time.
>
> A5: We respectfully note that we studied the extra computational costs of the classifier-based methods in Tables 7 and 8, using DVGO and NeuS for novel view synthesis and geometry reconstruction, respectively. The numerical results show that the extra computational costs of two components are marginal (only 6% for DVGO and 5% for NeuS).
>
>
> Reference:
>
> [1] Gao, Q., Xu, Q., Su, H., Neumann, U., & Xu, Z. (2023). Strivec: Sparse tri-vector radiance fields. In Proceedings of the IEEE/CVF International Conference on Computer Vision (pp. 17569-17579).

---

> > ### Comment · Reviewer_zn3A · 2023-11-22
> >
> > > Recent NeRF works, including Strivec (ICCV2023) and Zip-NeRF (ICCV2023), almost all aimed at (1) accurate and fast rendering, or (2) accurate rendering under some constrains (e.g. sparse/dynamic/corruption). Our experiments have covered these experimental settings.
> >
> > Thanks for the response. However, I didn't see a clear demonstration regarding the **fast speed** of the proposed method. Considering the recent methods targeting **accurate and fast**, merely reporting accuracy is insufficient. I would recommend including both training time and inference time comparisons with recent methods. Therefore, I will keep my score.

---

> > > ### Author Response · Authors · 2023-11-22
> > > **Further addressing the final concern on the training time and the inference time.**
> > >
> > > We sincerely thank for the reply.
> > >
> > > We would like to further address the reviewer's final concern on the training time and the inference time.
> > >
> > >
> > > Q6:  I didn't see a clear demonstration regarding the fast speed of the proposed method. Considering the recent methods targeting accurate and fast, merely reporting accuracy is insufficient. I would recommend including both training time and inference time comparisons with recent methods.
> > >
> > > A6: We would like to further include both training time and inference time comparisons with DVGO, NeuS, and Strivec. Both training time and inference time increase by 4-6 % for all of three neural field methods, which are marginal compared with the robust and significant performance improvement.
> > >
> > > ------------------------------------------
> > > | Model  | Training Time   | Inference Time |
> > > | ----------- | ----------- | ----------- |
> > > | DVGO-R  |  9.33 min     |      0.312 s |
> > >  DVGO-C  |  9.93 min     |      0.328 s
> > >  NeuS-R    |  2.84 h         |    10.27 s
> > >  NeuS-C   |   2.98 h       |      10.83 s
> > >  Strivec-R   |  1.16 h       |      4.38 s
> > >  Strivec-C   |  1.23 h        |     4.59 s
> > > ------------------------------------------
> > >
> > >
> > > In fact, the very marginal cost improvements fully match our expectation and is not surprising, because our method only modifies the number of logits of the final network layer. Such modification is very cheap, because it only happens at only one layer and does not slow any other processes. Note that classification loss, as a simple loss, obviously does not increase the inference time at all and matters little to the back-propagation time.
> > >
> > >
> > > We have updated the mentioned results in the revision.
> > >
> > > We present the quantitative and qualitative results of Strivec (ICCV 2023) in Table 12 and Figure 8, respectively. The results show that NFC may also improve recent regression-based neural field methods, while the extra computational cost are very marginal. We will try to present more results of recent SOTA methods in future.
> > >
> > > We would highly appreciate it, if the reviewer may re-evaluate our work.

---

> > > > ### Comment · Reviewer_zn3A · 2023-11-22
> > > >
> > > > Thanks for the explanation. This solves most of my concerns.
> > > >
> > > > I will raise my score to 6. For the final version, please add complementary experimental results regarding different methods (including DVGO, NeRF, NeuS, Strivec) on different datasets (including Replica Dataset, T&T Dataset). Table 1 and Table 2 show part of the combinations but not all of them. Also, I hope the algorithm can be open-sourced.

---

> > > > > ### Author Response · Authors · 2023-11-23
> > > > >
> > > > > We sincerely thank the reviewer for the constructive comments and kind support.
> > > > >
> > > > > We will try our best to meet your expectation.

---

### Author Response · Authors · 2023-11-22
**Paper Revision**

We sincerely appreciate all reviewers'  constructive comments and kind support!

We have revised our submission following the suggestions.

The revised contents are highlighted in red.

The mentioned supplementary results are presented Tables 7/8/12/13 and Figure 8.

- SOTA methods. We presented the quantitative and qualitative results of Strivec (ICCV 2023) in Table 12 and Figure 8, respectively. The results show that NFC may also improve very recent regression-based neural field methods. We will try to present more results of recent SOTA methods in future.

- Ablation study. We presented the ablation study results over 8 scenes in Table 13. The results show that the classification loss plays a dominant role in the proposed NFC, while the Target Encoding module is still often helpful is most scenes ($75%$) and lead to expected performance improvements. We also revised our discussion on ablation study and note the dominant role of the classification in Section 4.3 (red-colored parts).

- Training time and inference time. Please see Tables 7/8/12.

---

### Meta-Review · Area_Chair_XwAv · 2023-12-12

**Metareview:**

The paper studies whether regression learning in Neural Fields can be replaced by classification, which is an interesting question to begin with. To this end, the paper proposes a target encoding module combined with a classification loss, obtaining consistent improvements. Reviewers appreciate the straightforwardness and empirical validation of the work and have raised their scores during the rebuttal. As such, I recommend acceptance.

**Justification For Why Not Higher Score:**

It is unclear on whether there is a deeper theoretical substrate.

**Justification For Why Not Lower Score:**

Good results, clear idea.

---

### Decision · Program_Chairs · 2024-01-16

Accept (poster)